# Temperature and Host Fruit During Immature Development Shape Adult Life History Traits of Different *Ceratitis capitata* Populations

**DOI:** 10.3390/insects16010065

**Published:** 2025-01-11

**Authors:** Georgia D. Papadogiorgou, Nikos T. Papadopoulos

**Affiliations:** Laboratory of Entomology and Agricultural Zoology, Department of Agriculture, Crop Production and Rural Environment, School of Agricultural Sciences, University of Thessaly, Fytokou St., 38446 Volos, Greece; gewrgiapapadogi@yahoo.com

**Keywords:** Tephritidae, Diptera, life-history traits, demography, lifespan, overwintering hosts, invasion dynamics, range expansion

## Abstract

The Mediterranean fruit fly (medfly) is an extremely polyphagous pest, capable of infesting over 300 host plants, making it one of the most significant threats to fruit production globally. The aim of this study was to investigate how developmental temperature and host fruit type affect adult longevity and fecundity in medflies from six geographically distinct populations across Southern to Central Europe. The findings highlight the influence of environmental factors, such as temperature and host fruit, on the life history traits of *Ceratitis capitata* and underscore the species’ ability to adapt to different environments. This work offers important insights into how such adaptations contribute to the spread of this pest species and informs potential strategies for its management, especially in light of ongoing climate change.

## 1. Introduction

Environmental stress resistance and the ability to adapt to novel or changing environments are essential factors that influence the ecological patterns (e.g., population density, migration, habitat use) and biogeographic distribution of invasive species [1]. Invasive species can cause a range of negative impacts, including disruptions to ecosystems, loss of biodiversity, and economic effects in the invaded areas [2]. Climate change is expected not only to alter the insect distribution but also to increase the threat of invasive species as tropical and subtropical insects expand their geographic range into temperate areas [3,4]. Understanding the relationship between invasion and range expansion can provide insights into the mechanisms facilitating geographic spread [5].

Biological invasions, despite their detrimental ecological consequences, provide a valuable framework for studying evolutionary processes in real time. When non-native species establish in novel environments, they are exposed to selective pressures distinct from those in their native ranges. These pressures encompass a range of abiotic factors, such as climate and soil composition, as well as biotic interactions like competition and/or predation. To thrive, persist, and expand in these new habitats, invaders must rapidly adapt, which often results in accelerated evolutionary changes. These adaptations, driven by natural selection, phenotypic plasticity, and underlying genetic variation, enhance the species’ fitness in new conditions. The study of biological invasions, therefore, offers critical insights into evolutionary mechanisms, revealing how species respond and evolve under new environmental challenges beyond their native range [6].

The Mediterranean fruit fly (medfly), *Ceratitis capitata* (Wiedemann) (Diptera: Tephritidae), is a highly destructive pest with significant economic consequences for global fresh fruit production [7]. Having originated in Africa, *C. capitata* has rapidly expanded its distribution to temperate and tropical regions across the globe within the last two centuries [8]. In recent years, its presence has also been reported in cooler, more temperate zones [9,10]. The species’ successful invasion and establishment in these diverse environments can be attributed to local adaptation or phenotypic plasticity [3]. Several key biological and physiological characteristics of *C. capitata* contribute to its invasion success, including its high polyphagy, short life cycle, strong dispersal ability, and capacity for rapid adaptation to a wide range of climatic conditions [11,12,13,14]. Moreover, climate change is anticipated to further influence the global distribution and performance of *C. capitata* by creating conditions that enhance its survival and reproduction in areas previously unsuitable for its establishment [10,15].

Despite the extensive research on the biology and ecology of *C. capitata,* the extent to which physiological and life history traits contribute to its successful introduction into novel environments remains unclear [1]. Geographic diversity of genotypes and phenotypes in medfly populations can indeed emerge as a result of adaptation to novel habitats [16]. Responding to abiotic factors, like temperature, humidity, and photoperiod, as well as biotic factors such as host availability, can lead to adaptation that causes shifts in the genetic architecture of populations, resulting in alterations to crucial life history and behavioral traits [17]. Medfly biotypes originating from various regions worldwide, each with distinct invasion histories, have evolved different life history strategies to cope with local environmental conditions [6,18].

To comprehend the life history strategies that *C. capitata* may have developed to cope with local conditions at the front of its geographic range, Papadogiorgou et al. (2024) [19] examined the performance of immatures of six medfly populations that originated from environmentally diverse habitats (from Southern to Central Europe) to two key overwintering hosts (apples and bitter oranges) under a range of constant temperature conditions (15, 20, and 25 °C). Earlier studies demonstrated that medfly overwinters in cooler temperate areas as larvae in overwintering hosts (e.g., apples) that remain in orchards at relatively good conditions until spring [20,21,22]. The results revealed varying responses of medfly populations to different overwintering hosts and temperatures, highlighting the differential overwintering capacity of larvae of the different populations. Prevailing environmental conditions can stimulate developmental plasticity in individuals [23]. These plastic responses to environmental changes, including thermal stress, and the interaction between nutrition and thermal stress are considered important components of the *C. capitata* invasion success [14,24,25]. The rate of larval development is notably influenced by temperature and host fruit [26]. The effects of temperature encompass a wide range of physiological processes [27]. While warm temperatures tend to accelerate developmental rates, leading to a new generation consisting of poorly adapted individuals to winter conditions, lower temperatures may result in extreme phenotypes that possess adaptations allowing them to effectively cope with harmful winter conditions [21]. Similarly, host fruit cultivar in combination with ripening stage can influence the development of *C. capitata* immature stages and subsequently its demographic traits [28]. Larval diet with increased protein content tends to accelerate developmental processes, leading to shorter developmental time [23,29]. Hence the choice of nutritional resources (host selection) by adult females can influence medfly’s life history characteristics, such as the emergence of adults, pre-oviposition period, egg production, mating success, and adult longevity [30].

The demographic traits of the medfly have been extensively studied over the past few decades, with most research focusing on laboratory-adapted populations [26,31,32,33,34]. These populations, having undergone significant changes in life history traits through laboratory adaptation, may not accurately represent the natural life history strategies of geographically isolated medfly populations. In contrast, Diamantidis et al. (2008) [35] investigated the demographic responses of six geographically distinct medfly populations (from Greece, Kenya, Portugal, Brazil, Hawaii, and Guatemala) under controlled conditions. The study revealed that medflies from Kenya, Hawaii, and Guatemala exhibited shorter lifespans compared to those from Greece, Portugal, and Brazil. This finding suggests that regions with seasonal or temperate climates select for longer adult lifespans, likely as an adaptation to periods of limited host fruit availability.

Medfly populations established in regions with varying ecological characteristics demonstrated differences in several biological traits during both early developmental stages (larval/pupal) and adult life, reflecting local adaptations and the distinct selective pressures encountered in different environments. This is consistent with Malacrida et al. (1998) [36], who reported modest genetic divergence among medfly populations originating from the Mediterranean, compared to ancestral populations from Southeastern Africa. In tropical regions, where host fruits are more likely to be available throughout the year, adaptations may not favor longer adult lifespans. Thus, evolutionary processes that shape genetic traits may have had a limited impact on genes influencing life history traits, such as longevity [35]. However, little is known about the potential influence of immature developmental duration on the demographic characteristics of emerging adult medflies, warranting further investigation.

The aim of the present study was to investigate the impact of developmental plasticity on adult life history traits of geographically isolated *C. capitata* populations. Hence, we tested whether host and temperature during immature development affect demographic characteristics of adult medflies obtained across a latitudinal gradient of approximately 13 degrees. We tested the hypothesis that temperature and host during immature development have similar effects on both adult longevity and female reproduction in the different populations. We also hypothesize that the lower the developmental temperature, the longer the longevity of adults and the higher the fecundity of females as part of the plastic response.

## 2. Materials and Methods

### 2.1. Populations and Insect Rearing Conditions

The experiments were conducted in the laboratory of Entomology and Agricultural Zoology at the University of Thessaly from September 2019 to January 2021. The conditions inside the experimental rooms were set at 25 ± 1 °C, relative humidity of 45–55%, and a photoperiod of 14 h of light followed by 10 h of darkness. The light phase began at 07:00 a.m., and the lighting was provided by daylight fluorescent tubes with an intensity ranging from 1500 to 2000 lx. Six different medfly populations originating from Austria (Vienna), Croatia (Zaton), and Greece (Thessaloniki, Volos, Chios, and Crete) were used (Figure 1). To avoid altitudinal clines, the sampling sites covered a latitude range of approximately 35° to 48° N. To establish experimental colonies in the laboratory, approximately 1000 pupae from each sampling site were utilized. The population from Croatia originated from infested figs collected in Zaton (near Zadar), while the Austria population derived from infested pome fruits collected in Vienna and had been maintained in captivity for 1 to 2 generations. Three of the *C. capitata* populations from Greece, those from Volos, Chios, and Crete (Chania), were obtained from field-infested bitter oranges, while the population from Thessaloniki was from apples.

The colonies used in the present study consisted of approximately 100 adults per cage (wooden, wire-mesh cages, 30 × 30 × 30 cm) at 25 ± 1 °C, 45–55% relative humidity, and a photoperiod of 14 h of light followed by 10 h of darkness (photo phase started at 07:00 a.m.). Medflies had unlimited access to water and a standard adult diet consisting of yeast hydrolysate, sugar, and water in a 1:4:5 ratio. Females were allowed to lay eggs on red, hollow, punctured plastic domes with a diameter of 5.5 cm. Plastic domes were inserted into the lids of plastic petri dishes, with water added to maintain relative humidity. To trigger oviposition, a small cup containing 0.5 mL of fresh orange juice was placed in the base of a petri dish. Oviposition domes were left in the rearing cages for 24 h to collect the required number of eggs needed for the experiments. Before being used in our experiments, medflies originating from Zaton, Thessaloniki, Volos, and Chios were reared up to four generations (F1–F4), while flies from Crete were reared up to F8–F9 generations. Finally, medfly populations originating from Vienna were reared under laboratory conditions for F9 generations.

### 2.2. Host Fruits

The pre-adult survival and development of different medfly populations were assessed within the two key overwintering hosts (apples and bitter oranges) [21,37]. Apples (Golden delicious) and bitter oranges (local cultivar) were collected from organic orchards located in Naoussa, Northern Greece, and Lechonia (Volos), respectively. Upon collection, all fruits were washed with tap water and stored at a temperature of 6 ± 2 °C before being used in an experiment.

### 2.3. Development of Immature Stages

The methodology for egg implantation and subsequent development of *C. capitata* in host fruits is given in detail in Papadogiorgou et al. (2024) [19], study. Briefly, freshly laid eggs (24 h) were obtained from the oviposition domes and used for implantation into the two host fruits. In each fruit, two artificial holes (1 mm in diameter) were drilled on opposite sides of the upper part, and 5 eggs were artificially implanted into each hole (10 eggs/fruit). The infested fruits were then individually placed in plastic containers on a layer of sterilized sand, covered with organdie cloth, and transferred to the three constant temperatures (15, 20, and 25 °C). An inspection of artificially infested fruits was conducted daily. Newly formed pupae were carefully recorded, collected, and maintained in the same thermal conditions until adult emergence. This experimental setup is consistent with the approach outlined in Papadogiorgou et al. (2024) [19], where detailed information on the immature stages and rearing conditions can be found. Particularly, for each population, we conducted 20 replications per temperature/host combination (1 fruit with 10 eggs), resulting in a total of 360 apples and 360 bitter oranges (20 fruits per population and temperature for the 6 medfly populations).

### 2.4. Adult Demography

All adults that emerged from each treatment were utilized to investigate the effects of population-origin, host, and temperature during pre-adult development on adult longevity and female fecundity. For each combination of origin, host, and temperature, all available adults were used to form pairs (1 female and 1 male). Each pair of adults was placed into an individual cage (400 mL plastic cup) and transferred to 25 ± 1 °C, 45–55% relative humidity, and a 14:10 L:D photoperiod. To assist egg collection, a dome (5 cm) at the base of each cup was used. The dome was fitted into a 5 cm hole made on the cover of a plastic petri dish. To maintain humidity levels beneath the dome, adequate water was added to the base of the petri dish. Adults had access to water and food consisting of yeast hydrolysate, sugar, and water in a ratio of 1:4:5, respectively. Adult mortality and female fecundity were recorded daily until death. The number of adults that emerged varied across treatments, which determined the number of replicates (pairs) used for each combination of population, host, and temperature (Appendix A).

### 2.5. Data Analysis

Generalized Linear Models (GLMs) were applied to evaluate the effects of host, population, temperature, and sex on lifespan. For fecundity and the durations of pre-oviposition, oviposition, and post-oviposition periods, only females were included in the analysis, with host, population, and temperature as explanatory factors. Lifespan and the reproductive periods (pre-oviposition, oviposition, and post-oviposition) were modeled using a linear regression. Fecundity, characterized by count data with overdispersion, was modeled using a negative binomial regression with a log link function. Pairwise comparisons were adjusted by using the Bonferroni correction test.

## 3. Results

### 3.1. Effect of Host Fruit and Developmental Temperature on Lifespan of Ceratitis capitata Adults

In Figure 2, the adult lifespan of six distinct medfly populations is presented. Adults of the different populations obtained from larvae that had developed in either apples or bitter oranges under three constant temperatures (15, 20, and 25 °C). Neither host fruit (apple or bitter orange) nor sex had a significant effect on adult lifespan (*p* = 0.115; 0.332, respectively) (Table 1). However, developmental temperature influenced adult longevity, with higher temperatures generally associated with an increase in lifespan (*p* < 0.001). Significant differences in adult lifespan were observed across populations, regardless of the host fruit or developmental temperature to which larvae were exposed (*p* < 0.001). Flies from Zaton exhibited the longest lifespan. Longevity was similar among adults from Vienna (*p* > 0.05), Thessaloniki (*p* > 0.05), and Crete (*p* > 0.05) (Appendix A). The interactions between population and temperature (*p* < 0.001) as well as population and sex (*p* < 0.001) were significant, suggesting that the developmental temperature differentially affects adult longevity across medfly populations (Table 1). Higher developmental temperatures generally led to longer lifespans, but the extent of this effect differed between populations. The increase in temperature in adult lifespan did not differ among flies from Crete (baseline), Zaton, Volos, and Chios (*p* > 0.05). However, the increase in temperature affected adult lifespan in flies from Vienna and Thessaloniki compared to flies from Crete (baseline) (*p* = 0.036; 0.009, respectively) (Appendix A). In all tested populations, males outlived females (*p* < 0.001), with the exception of the populations from Volos and Crete, where lifespan among males and females was comparable (*p* > 0.05) (Appendix A).

### 3.2. Effect of Host Fruit and Developmental Temperature on Fecundity of Ceratitis capitata Females

The average fecundity of ovipositing females at 25 °C for six distinct populations, derived from immatures reared on either apples or bitter oranges at three constant temperatures, are presented in Figure 3. Fecundity was significantly influenced by temperature, with females that developed at 25 °C as immatures laying more eggs than those reared at lower temperatures (*p* = 0.009), regardless of population or host. No significant differences in fecundity were observed between the two host plants (*p* > 0.05). Similarly, population origin did not have a significant effect on fecundity (*p* > 0.05) (Table 2).

### 3.3. Effect of Host Fruit and Developmental Temperature on Female Reproductive Periods

The reproductive periods (pre-oviposition, oviposition, and post-oviposition) for females of *C. capitata* from different populations reared on apples and bitter oranges at 15, 20, or 25 °C as immatures are shown in Figure 4. The pre-oviposition period was affected by both the host fruit and the population (*p* < 0.001), but not by temperature (*p* > 0.05) (Table 3). In general, females reared on apples had a longer pre-oviposition period than those reared on bitter oranges. Among populations, females from Thessaloniki had the longest pre-oviposition period (Appendix A). The interaction between host and population was a significant predictor of pre-oviposition period (*p* < 0.001) (Table 3). Females from Thessaloniki (*p* < 0.001) and Volos (*p* = 0.020) had longer pre-oviposition periods when reared as immatures on apples compared to those obtained from bitter oranges. In contrast, females from Vienna, Zaton, Chios, and Crete (*p* > 0.05) showed no significant difference in pre-oviposition periods between the two host fruits (Appendix A).

The oviposition period was significantly affected by host (*p* = 0.040) and population (*p* < 0.001), but not by the temperature at which immatures have developed (*p* > 0.05) (Table 3). Specifically, the oviposition period was longer in bitter oranges than in apples, irrespective of the tested population and temperature. Females from Zaton exhibited the longest oviposition period, which was comparable to that of females from Vienna (*p* > 0.05) and Crete (*p* > 0.05) (Appendix A). Additionally, although temperature alone had no significant effects on the oviposition period, the interaction between population and temperature had (*p* = 0.004). Hence, some populations experienced a longer oviposition period as temperature increased, while others showed the opposite trend. In Crete (baseline), the increase in temperature on oviposition period was similar to that observed in Vienna and Zaton (*p* > 0.05) (Appendix A).

The data presented in Figure 4 demonstrate that the post-oviposition period was influenced by all tested factors, including host, population, and temperature (Table 3). Across all populations, the post-oviposition period was marginally longer in females reared on apples compared to those reared on bitter oranges, regardless of the temperature during immature development (*p* = 0.046). Females from Crete exhibited the longest post-oviposition period compared to all other populations, which was similar to the post-oviposition period for females originating from Thessaloniki (*p* > 0.05) (Appendix A). Additionally, an increase in temperature resulted in an extended post-oviposition period in females, regardless of population (*p* < 0.001). The interaction between host and population also significantly affected the post-oviposition period (*p* < 0.001). In all tested populations, the post-oviposition period was similar among the two hosts, with the exception in Crete, where females reared on apples had a notably extended post-oviposition period (*p* < 0.001) (Appendix A). Furthermore, the interaction between population and temperature was also a significant predictor of post-oviposition period (*p* < 0.001). In all tested populations, higher temperatures were associated with a reduction in post-oviposition period (Appendix A).

The proportion of the total female lifespan allocated to the pre-oviposition, oviposition, and post-oviposition periods was calculated for each population, host fruit, and temperature by dividing the duration of each stage by the total lifespan of each individual (Figure 5). Medflies from Thessaloniki, Volos, and Chios exhibited significantly longer pre-oviposition periods as a proportion of their total lifespan compared to other populations (*p* < 0.001). In these populations, residual analysis revealed that the pre-oviposition period accounted for approximately 42% of the total lifespan, whereas in most other populations, it represented only 15–19%, with the exception of the Vienna population (33%). Similarly, the oviposition period as a proportion of the total lifespan was significantly longer in medflies from Zaton (*p* < 0.001); however, when analyzed in terms of average chronological duration (Figure 3), no significant difference was observed between the oviposition periods of the Zaton and Crete populations (*p* > 0.05). Medflies from Crete exhibited the longest post-oviposition period, both in terms of chronological duration (Figure 4) and as a proportion of the total lifespan (Figure 5).

Between the two tested hosts, the pre-oviposition period was longer in terms of chronological duration for medflies reared on apples (Figure 4). However, when expressed as a proportion of the total lifespan, the pre-oviposition period was comparable between the two hosts (*p* > 0.05). The oviposition period was significantly longer for medflies reared on bitter oranges, both in absolute duration and as a proportion of the total lifespan (*p* < 0.001). In contrast, the post-oviposition period was longer for medflies reared on apples in both metrics (*p* = 0.005). Finally, temperature influenced the post-oviposition period similarly across all populations, with consistent effects observed in both absolute and proportional terms.

## 4. Discussion 

Our findings demonstrate that developmental temperature regulated both adult lifespan and female reproductive traits in *C. capitata*, while the effects of host fruit and population were weaker. Higher temperatures during development were generally associated with increased adult longevity and egg production, with significant effects across populations. The Zaton population exhibited the longest lifespan and highest egg production, with notable interactions between temperature and population. Host fruit differently affected reproductive periods across populations, with apples generally extending the pre-oviposition period and bitter oranges lengthening the oviposition period. Additionally, the post-oviposition period was influenced by all tested factors, including host, population, and temperature, underscoring the complex interactions that shape *C. capitata* reproductive dynamics. These findings highlight the importance of environmental conditions and genetic variability in shaping medfly life history traits. They also underscore the ability of different medfly populations to generate a long list of phenotypes in response to both biotic and abiotic conditions (e.g., host fruit and temperature, respectively).

### 4.1. Effect of Host Fruit and Developmental Temperature on Lifespan of Ceratitis capitata Adults

The quality and quantity of nutrients acquired during larval development are critical determinants of insect fitness, impacting reproduction, aging, and stress tolerance in adulthood [38,39,40,41]. In holometabolous insects like *C. capitata*, nutrient reserves accumulated during the larval stage play a pivotal role in supporting the non-feeding pupal stage and sustaining adult energy demands [42,43]. As demonstrated earlier, the nutritional quality of larval host fruits significantly influences adult longevity and fecundity [44]. Nutrient-dense fruits, such as peaches and persimmons, enhance adult lifespan, likely due to their balanced provision of macronutrients essential for cellular maintenance and energy storage. Conversely, nutrient-poor hosts, such as apples and pears, result in adults with limited energy reserves and reduced reproductive potential [45].

In the present study, no significant differences were observed in the adult longevity of *C. capitata* individuals reared on two key overwintering hosts, despite prior evidence of host-dependent effects on immature survival and development time [19]. This finding may reflect the standardized conditions provided during adulthood, including a constant temperature of 25 °C, a nutritionally optimal adult diet, and equal mating opportunities. Such conditions likely minimized the carry-over effects of larval nutrition, neutralizing the influence of host fruit on adult lifespan. This suggests that, while larval diet is critical for immature survival and development, its impact on adult longevity may be less pronounced under controlled and stable environmental conditions and optimal feeding resources.

Temperature is a key abiotic factor influencing insect development, survival, and reproductive success [46]. Lower developmental temperatures have been shown to extend larval periods, allowing greater nutrient accumulation, even from suboptimal diets [28,47,48]. However, our study revealed the opposite. This deviation from expectations may be explained by the warm-climate origin of most experimental populations and the higher tested temperature of 25 °C, which falls within the optimal range for *C. capitata*. At this temperature, metabolic efficiency may facilitate energy conservation, enhancing adult survival [49].

Geographic variation in the lifespan of *C. capitata* reflects the interplay of genetic differentiation, local adaptation, and invasion history. Northern populations tend to exhibit longer lifespans, which aligns with life-history strategies typical of temperate regions, where extended longevity is beneficial for overwintering during adverse conditions [18]. In a previous study, Papadogiorgou et al. (2024) [19] found that populations from southern, warmer regions of Europe exhibited a longer egg-to-adult developmental duration compared to populations from northern, cooler regions. Based on these findings, our hypothesis for the current study was that populations from warmer latitudes (i.e., lower latitudes) would have shorter lifespans due to faster, more accelerated development. Our results supported this hypothesis. However, the Crete population showed a lifespan similar to populations from cooler regions like Vienna, Zaton, and Thessaloniki.

The observed longer lifespan in northern populations of *C. capitata* aligns with life-history strategies typical of temperate regions, where extended longevity is crucial for surviving harsh winter conditions. This finding suggests that northern populations may have adapted to cope with cooler climates through extended lifespans, which could enhance their survival and reproductive success during periods of environmental adversity. Overwintering strategies play a significant role in this adaptation. For example, as demonstrated by Wernicke et al. (2024) [50], *C. capitata* in Vienna is unable to survive the winter outdoors in the open field but can endure in protected environments, such as basements, where more stable conditions prevail. This ability to overwinter in urban microclimates, aided by the urban heat island effect [51,52], suggests that northern populations may rely on similar strategies to survive colder months, thereby supporting longer lifespans. Similarly, in Central Dalmatia, *C. capitata* overwinters as pupae in natural settings or as adults in urban environments [53]. By contrast, in Mediterranean regions like Thessaloniki, Volos, and Chios, overwintering predominantly occurs in the larval stage within fruits, such as apples or bitter oranges [21,54]. The Crete population’s extended lifespan may reflect its flexibility to overwinter across multiple developmental stages, providing adaptive advantages in response to fluctuating environmental pressures [37].

Sexual dimorphism in lifespan was evident, with males consistently outliving females across all populations, consistent with previous findings [18,55]. This pattern likely arises from the differing reproductive roles of each sex. Female lifespan is constrained by the physiological costs of egg production, which accelerate aging and reduce survival [56]. In contrast, males experience lower reproductive costs, primarily associated with mating, which may contribute to longer longevity.

### 4.2. Effect of Host Fruit and Developmental Temperature on Fecundity of Ceratitis capitata Females 

Our initial hypothesis—that lower developmental temperatures during *C. capitata*’s immature stages would lead to longer lifespans and higher fecundity in females, assuming slower development at lower temperatures allows for greater nutrient accumulation and extended adult longevity—was rejected [57]. Overall, the higher the temperature during immature development, the longer the adult lifespan. However, this pattern is not consistent across all tested populations and hosts. This finding suggests that life history traits in *C. capitata* may be influenced by the thermal history during the immature stages, as suggested by other studies on insect development and fitness [26,48]. Accelerated development at high temperatures allows females to initiate oviposition earlier, which may align their reproductive output with optimal environmental conditions and available resources [23,58]. While we provided the test flies with unlimited food and water to control resource availability, we acknowledge that periods of food or water stress could yield different results. However, testing the effects of adult food availability and/or quality was beyond the scope of our current study. Future studies could explore the potential interaction between immature development and adult food availability in shaping adult life history traits. Furthermore, temperature is known to modulate the synthesis of juvenile hormone (JH), which plays a crucial role in regulating molting, reproductive development, and other physiological processes in insects. Elevated temperatures during the immature stages can enhance JH synthesis, which in turn accelerates oocyte maturation and increases reproductive output by enabling females to produce eggs more rapidly [59,60].

### 4.3. Effect of Host Fruit and Developmental Temperature on Female Reproductive Periods

#### 4.3.1. Pre-Oviposition Period

Females reared on apples exhibited longer pre oviposition periods compared to those on bitter oranges. This supports previous findings linking bitter oranges to faster reproductive maturation, likely due to higher nitrogen availability and lower inhibitory compounds [61]. Prolonged larval development, as observed in immatures reared on apples [19], delays reproduction by shifting energy allocation toward development and survival, a phenomenon previously reported in other insect systems [62].

The prolonged pre-oviposition period in the Thessaloniki population may represent an adaptive delay to synchronize reproduction with favorable conditions, in line with previous findings of high developmental plasticity in this population during immature stages [19]. Such reproductive delays likely provide survival advantages in temperate regions, where overwintering plays a critical role in population persistence [21]. Conversely, the shorter pre-oviposition periods observed in the Zaton population may reflect evolutionary constraints. As a recently established population on the Northern Adriatic coast, limited genetic diversity resulting from founder effects could reduce their capacity for adaptive plasticity, thereby constraining their ability to adjust reproductive timing to environmental variability [13,53].

The interaction between population origin and host fruit further supports this interpretation. Populations from Thessaloniki, Volos, and Chios had longer pre-oviposition periods on apples, while those from Vienna, Zaton, and Crete showed no significant differences between apples and bitter oranges, suggesting a more generalized or stable reproductive strategy in the latter populations. For example, Crete’s warm climate and year-round availability of host fruits might reduce the selective pressure for host-specific adaptations, while the newer establishment of *C. capitata* in Zaton [53] may limit its ability to exploit a wide variety of hosts effectively, possibly due to reduced genetic diversity [13].

#### 4.3.2. Oviposition Period

Females reared on bitter oranges exhibited longer oviposition periods, likely due to the higher nutritional quality of this host fruit, which provided better larval development conditions. As reported by Kapsi et al. (2002) [48], nutrient-rich diets can enhance larval development and increase nutritional reserves, supporting sustained reproductive output. Conversely, apples, characterized by lower nitrogen content and potentially deterrent compounds, likely experienced slower larval development [19] and resulted in reduced reproductive capacity, as evidenced by the shorter oviposition periods.

The relationship between lifespan and reproductive timing underscores population-specific life-history strategies. We hypothesized that populations from cooler regions would exhibit longer lifespans and extended reproductive periods. Our results support this hypothesis. This pattern is consistent with the hypothesis that variations in lifespan influence reproductive schedules, as suggested by Diamantidis et al. (2009) [18]. These findings align with the idea that reproductive strategies in geographically distinct populations are shaped by differences in survival patterns and local environmental pressures [63]. Müller et al. (2009) [63] proposed that population-specific survival schedules are predictive of individual reproductive trajectories, a concept reflected in our results. The observed differences in lifespan among *C. capitata* populations appear to influence the timing and duration of oviposition, indicating that reproductive timing variation is likely an adaptive response to population-specific survival constraints rather than intrinsic differences in reproductive potential.

#### 4.3.3. Post-Oviposition Period

Females from Crete exhibited the longest post-oviposition periods, potentially reflecting adaptations to Crete’s unique agroecosystems, where fruit availability and predation pressures may select for extended lifespan strategies. This observation aligns with Diamantidis et al. (2009) [18], who suggested that post-reproductive females may act as ecological decoys, diverting predation from younger, actively reproducing individuals.

The selective pressures driving variation in post-reproductive lifespan remain unclear. While ecological drivers likely shape post-reproductive lifespan in *C. capitata*, studies in other species, such as guppies, suggest this phase may result from neutral processes without direct fitness benefits [64]. This contrast underscores the complexity of understanding post-reproductive lifespan, warranting further investigation into its ecological and evolutionary determinants.

Although the type of host fruit alone had only a marginal influence on the duration of the post-oviposition phase, the combination of host fruit and population origin played a substantial role in shaping this life-history trait. Most populations had similar post-oviposition durations regardless of host fruit, consistent with previous studies [57]. However, Crete individuals showed a significantly longer post-oviposition period on apples compared to other populations. The prolonged post-oviposition period observed in the Crete population of medflies likely reflects local adaptations to the region’s warmer climate and the continuous availability of diverse host fruits, as suggested by Mavrikakis et al., (2000) [37]. These environmental factors could drive evolutionary changes that enhance the medflies’ ability to survive and reproduce efficiently in Crete’s unique ecological conditions. Specifically, the extended post-oviposition period might serve as a strategy that allows medflies to maximize reproductive success and longevity under specific regional stresses. Further research is needed to uncover the genetic, physiological, and ecological mechanisms that contribute to these population-specific adaptations.

## 5. Conclusions

This study highlights the pivotal role of developmental conditions during the immature stages in shaping the life history traits of *C. capitata*, emphasizing the species’ remarkable plasticity and population-specific adaptations to diverse environmental conditions. The observed geographic variation in traits such as longevity, fecundity, and responses to developmental temperatures underscores the species’ ability to adjust its life history strategies to optimize survival and reproduction across a range of climates. This adaptability is a key driver of *C. capitata*’s invasive potential, allowing it to exploit favorable conditions created by global warming, such as extended fruiting seasons and increased availability of tropical and subtropical fruits in cooler regions. Even modest increases in average temperatures can reduce the severity of cold stress, enhancing survival rates in temperate areas and enabling population establishment in previously unsuitable habitats. These findings provide crucial insights into the mechanisms underpinning the species’ success as a global pest, offering valuable predictive tools for assessing its future range expansion under climate change. Moreover, this study underscores the importance of incorporating population-specific traits and environmental variability into the design of pest management strategies, ensuring they are tailored to local conditions to effectively mitigate the impacts of this highly adaptable and invasive species.

## Figures and Tables

**Figure 1 insects-16-00065-f001:**
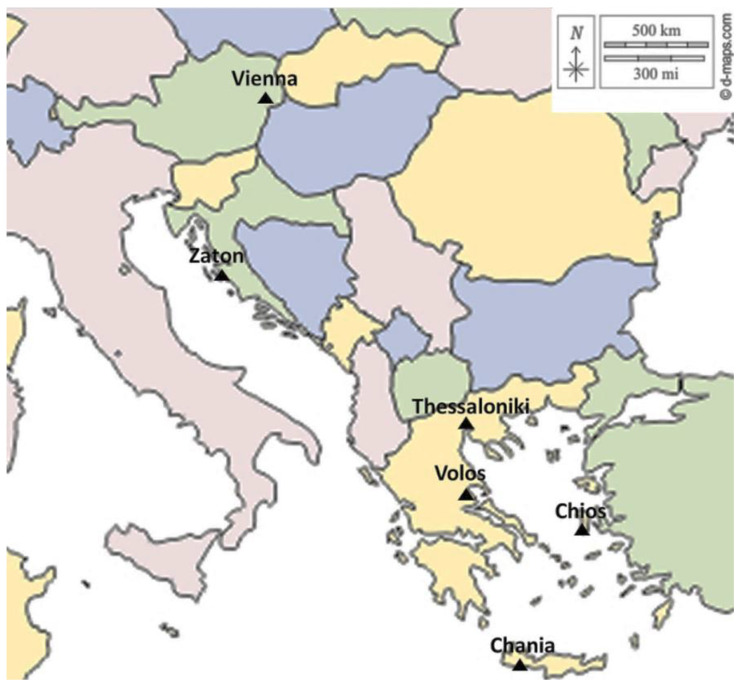
Locations across a latitudinal cline in East Mediterranean and Central Europe, where *Ceratitis capitata* populations originated. The map template obtained from https://d-maps.com (accessed on 20 October 2024).

**Figure 2 insects-16-00065-f002:**
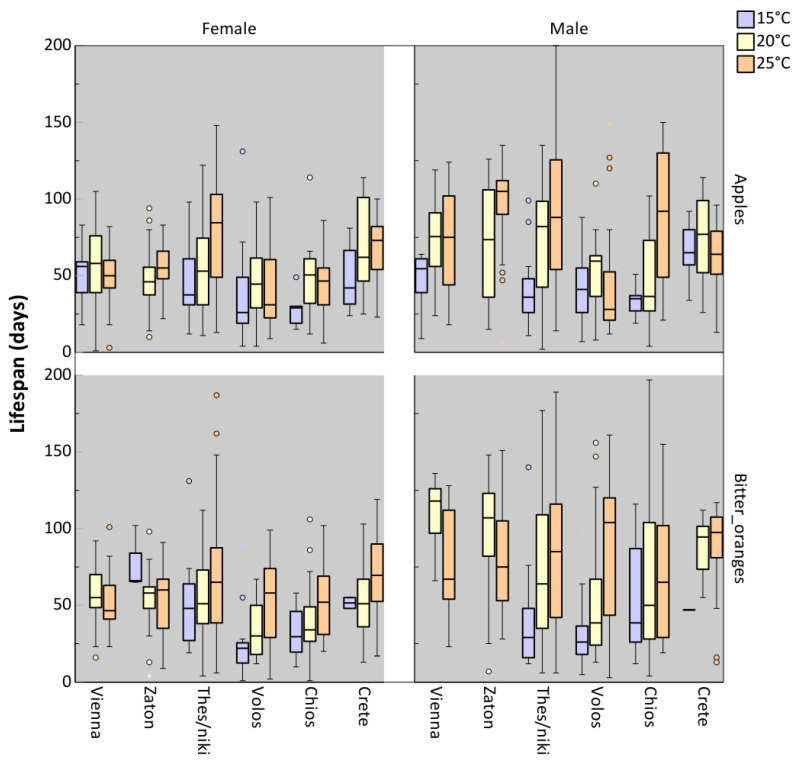
Lifespan of females and males of different *Ceratitis capitata* populations, derived from larvae developed in apples and bitter oranges at three constant temperatures (15, 20, and 25 °C). Boxplots include the median, the 1st and the 3rd quartile. Whiskers indicate the lowest-highest value inside the interval defined by ±1.5-fold interquartile range from the 1st/3rd quartile. Circles represent outliers, which are data points outside this interval.

**Figure 3 insects-16-00065-f003:**
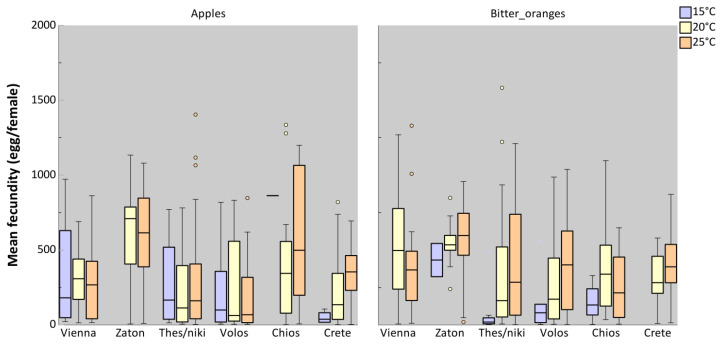
Boxplots of lifetime fecundity rates of females of different *Ceratitis capitata* populations obtained from apples and bitter oranges kept at 15, 20, and 25 °C, respectively. Boxplots include the median, the 1st and the 3rd quartile. Whiskers indicate the lowest-highest value inside the interval defined by ±1.5-fold interquartile range from the 1st/3rd quartile. Circles represent outliers, which are data points outside this interval.

**Figure 4 insects-16-00065-f004:**
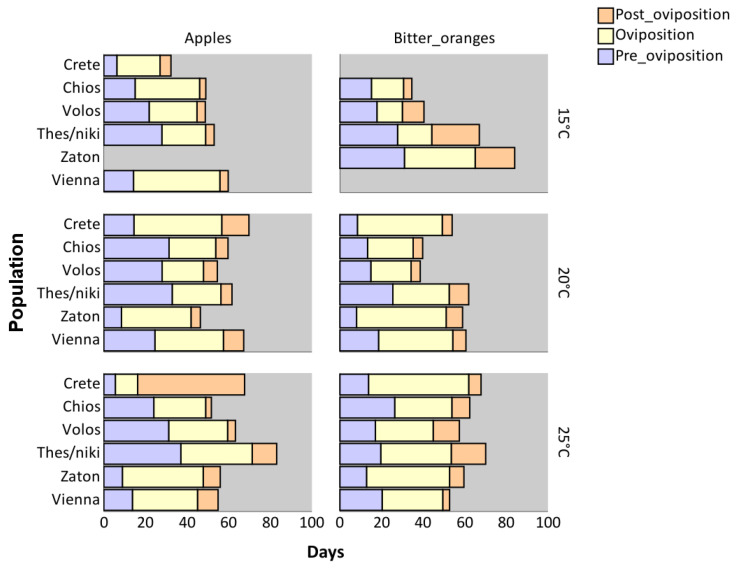
Reproductive periods (pre-oviposition, oviposition, post-oviposition period) for females from different *Ceratitis capitata* populations reared in apples and bitter oranges under different constant temperatures.

**Figure 5 insects-16-00065-f005:**
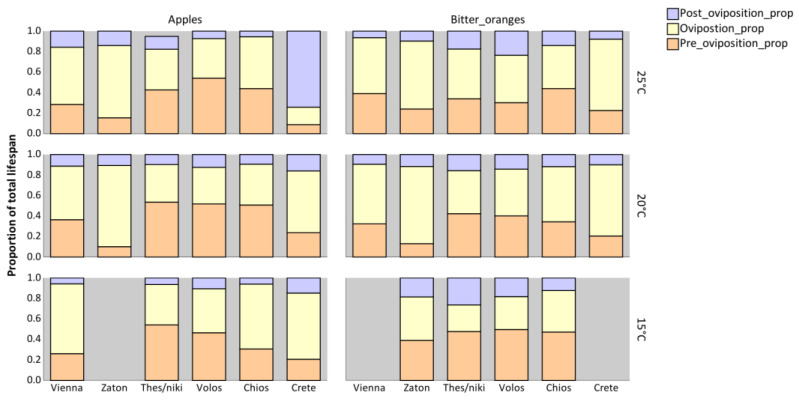
Proportion of total lifespan spent in pre-oviposition, oviposition, and post-oviposition activities for the different *Ceratitis capitata* populations reared in apples and bitter oranges under different constant temperatures.

**Table 1 insects-16-00065-t001:** Results of the Generalized Linear Models (GLMs) testing the effects of several variables on the lifespan of *Ceratitis capitata* adults from different populations reared on apples and bitter oranges at 15, 20, and 25 °C.

Variables in the Model	Wald *x*^2^	df	*p*
Intercept	37.698	1	<0.001
Host	2.748	1	0.115
Population	38.151	5	<0.001
Sex	0.940	1	0.332
Temperature	25.652	1	<0.001
Population x Sex	26.989	5	<0.001
Population x Temperature	33.606	5	<0.001
Sex x Temperature	1.100	1	0.294

**Table 2 insects-16-00065-t002:** Results of Generalized linear models testing the effects of population, host, and temperature treatment on the fecundity of *Ceratitis capitata* females.

Variables in the Model	Wald *x*^2^	df	*p*
Intercept	261.511	1	<0.001
Host	2.756	1	0.097
Population	9.342	5	0.096
Temperature	6.902	1	0.009
Population x Temperature	7.356	5	0.195

**Table 3 insects-16-00065-t003:** Results of Generalized linear models testing the effects of variables on the duration of the pre-oviposition, oviposition, and post-oviposition periods of *Ceratitis capitata* females from different populations reared as immatures on apples and bitter oranges at 15, 20, and 25 °C.

Variables in the Model	Wald *x*^2^	df	*p*
Pre oviposition period			
Intercept	17.185	1	<0.001
Host	15.273	1	<0.001
Population	175.784	5	<0.001
Temperature	0.495	1	0.482
Host x Population	37.731	5	<0.001
Oviposition period			
Intercept	18.639	1	<0.001
Host	4.216	1	0.040
Population	23.110	5	<0.001
Temperature	1.230	1	0.267
Population x Temperature	17.534	5	0.004
Post oviposition period			
Intercept	4.808	1	0.028
Host	3.967	1	0.046
Population	33.267	5	<0.001
Temperature	17.926	1	<0.001
Host x Population	91.437	5	<0.001
Host x Temperature	5.987	1	0.014
Population x Temperature	43.337	5	<0.001

## Data Availability

Data will be made available on request.

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
