# Peer review of "Temperature and Host Fruit During Immature Development Shape Adult Life History Traits of Different Ceratitis capitata Populations"

_insects, 2025, doi:10.3390/insects16010065_

Round 1
Reviewer 1 Report
Comments and Suggestions for Authors
Dear Editor,
I read the manuscript by Papadogiorgou & Papadopoulos entitled "Temperature and Host Fruit During Immature Development Shape Adult Life History Traits of Different Ceratitis capitata populations". The manuscript describes the effect of different temperature regimes during the larval stage, the host fruit, and the population of origin on the longevity and fecundity of C. capitata. The question addressed is both interesting and relevant considering the importance of this species as a fruit pest; the experiments were carefully carried out; results were well-analyzed and presented and the manuscript is well written. I Have only detected minor details that should be considered before publication.
COMMENTS
L23: mention the three temperatures used
L135: check the text
L157: delete "and"
L169: Host Fruits
L170-171: please add a supporting reference
L204: unclear, sex a fixed factor to analyze fecundity and durations of pre-post oviposition?
L227: "The effect of temperature..." here and elsewhere try to be more explicit (the positive/negative effect, increase/decrease, etc)
L229: move "(P = 0.036; 0.009, respectively)" after the populations are named.
L240: change "adults" to "females"
L331: change to "female reproductive traits".
L332: I do not think the word "stochastic" fits well here. Maybe "less clear" or "weaker"...
L370: "our study demonstrated that higher temperatures during develop-369 ment were associated with shorter larval durations.." how is written gives the impression this is part of the current paper. Please modify the writing.
L380: "warmer regions" of Europe?
L416-418: I understand that statistical results indicate this, but the figure shows that this pattern is not clear at all from apples.
L424-428: It should be mentioned that these findings were not specifically reported for C. capitata or any other related tephritid.
General: the discussion is rather long. Try to reduce it (for instance, avoid repetition of results in the first sentences of each section)
Author Response
- L23: mention the three temperatures used
Response: Thank you for your suggestion. In the revised text, I have included the three temperatures used for clarity (line: 23)
- L135: check the text
Response: Thank you for pointing that out. We have reviewed the sentence and revised it for clarity and accuracy (line 135)
- L157: delete "and"
Response: Thank you for pointing that out. In the revised text we have deleted “and” (line: 158), also we changed a bit the text as a suggestion of another reviewer
- L169: Host Fruits
Response: Following reviewers comment we changed the subheading (line 171)
- L170-171: please add a supporting reference
Response: Thank you for your suggestion. We have included references relevant to the two key overwintering hosts in the revised text (line 173).
- L204: unclear, sex a fixed factor to analyze fecundity and durations of pre-post oviposition?
Response: Thank you for your comment. Only females were used for the analysis of fecundity and the durations of pre- and post-oviposition periods (lines: 210-212).
- L227: "The effect of temperature..." here and elsewhere try to be more explicit (the positive/negative effect, increase/decrease, etc)
Response: Thank you for your valuable feedback. We have addressed the comment by explicitly describing the direction of the effects of temperature (lines: 234,235, 287)
- L229: move "(P = 0.036; 0.009, respectively)" after the populations are named.
Response: Following reviewers’ comment, the P values were mentioned after the populations in the revised text (line 237)
- L240: change "adults" to "females"
Response: Thank you for pointing that out. In the revised text we included females (line 249).
- L331: change to "female reproductive traits".
Response: Following reviewers comment, we added the word “female” (line 341)
- L332: I do not think the word "stochastic" fits well here. Maybe "less clear" or "weaker".
Response: Following reviewers comment, in the revised text we added less clear instead of stochastic (line 342).
- L370: "our study demonstrated that higher temperatures during develop-369 ment were associated with shorter larval durations.." how is written gives the impression this is part of the current paper. Please modify the writing.
Response: Thank you for your comment. In the revised text, we deleted this phrase accordingly to you last comment to avoid repetitions of our results
- L380: "warmer regions" of Europe?
Response: Following reviewers comment we specifies that we referred to warmer regions of Europe (line 388).
- L416-418: I understand that statistical results indicate this, but the figure shows that this pattern is not clear at all from apples.
Response: Thank you for your comment. The apparent lack of a clear pattern for some populations in apples is now discussed see “421-423”. The discussion now better reflects the results of the statistical analysis that show a significant effect of temperature on adult lifespan and fecundity, and discuss the observed variability.
- L424-428: It should be mentioned that these findings were not specifically reported for C. capitata or any other related tephritid.
Response: Thank you for your suggestion. We have added a reference regarding a study on Bactrocera cucurbitae, which indicates that with an increase in temperature, juvenile hormone (JH) levels were also increased. While these findings were not specifically reported for C. capitata, the referenced study provides relevant context. You can see the reference added in line 437.
- General: the discussion is rather long. Try to reduce it (for instance, avoid repetition of results in the first sentences of each section)
Response: In the revised text, we tried to address the comment by reducing the length of the discussion and avoiding the repetition of results in the opening sentences of each section.
Reviewer 2 Report
Comments and Suggestions for Authors
The paper ‘Temperature and host fruit during immature development shape adult life history traits of different Ceratitis capitata populations’ is a solid contribution that investigates the impact of population, larval substrate (host fruit), and temperature of immature development on adult demography, including lifespan, fecundity, and durations of female reproductive phases (oviposition period plus pre- and post-oviposition). Importantly, this investigation quantifies the degree to which developmental conditions influence adult demography among different populations and so demonstrates the population-level adaptations to different biotic and abiotic environmental conditions (both biotic and abiotic).
Line-by-line comments are given below. Many of these are minor editorial comments, but I suggest the authors clarify issues of sample size (lines 186-196, line 201, line 213).
Line 37 – delete a
38 – what are ecological patterns? Use more precise language.
40 – why both ecological and environmental effects? How can such effects be distinguished. Re-word.
41 – delete the; change insects to insect
58 – insert comma after closing parenthesis; delete the before global
59 – change to Having originated in Africa, …
70- replace at with to
71 – delete comma after traits
72 – change remain to remains [singular to agree with extent]
74 – insert comma after factors
81 – delete (2024)?
84 – delete comma [after …bitter oranges)]
86 – remain in orchards [not on]
87 – delete plastic
91 – insert comma after changes
92 – insert comma after stress
92 – delete as
103 – change to ..medfly’s life history characteristics, such as…
120 – delete (1998)?
122 – change to …regions, where host fruits are more likely to be available throughout the year…suggesting constancy in fruit availability is perhaps unwarranted.
125 – insert comma after traits
133 – change effect to effects
135 – what is a fecundity rate? Just use fecundity. Two hanging words = ‘plastic response’ – delete.
145 – insert and before Greece
146 – not sure what phrase ‘To avoid altitudinal clines’ means. Perhaps delete this.
148 – change to The population from Croatia…
150 – change to ...had been maintained…
151-152 – Three of the C. capitata populations from Greece, those from Volos, Chios, and Chania, developed in oranges, while the population from Thessaloniki developed in apples.
156-157 – change to …100 adults per cage (wooden, wire-mesh, 30x30x30 cm) at 25 + ….
160 – delete mixture; so I understand – all ingredients were mixed together?
163 – change to trigger
165-168 – the generation numbers given refer to generations of rearing in lab prior to use in the experiments, correct? Perhaps state that explicitly.
2.2 – reverse order- use Host Fruits as section heading
178 – delete (2024)?
180 – delete comma after closing parenthesis
184 – how many fruit of each type were placed in each of the 3 temperature regimes?
184 – change to artificially
186-189 – put sample size information above; delete sentence starting This experimental setup…
193-196 – Do I understand correctly? Only 1 pair of flies was used per origin/host/temperature? As female fecundity can vary a lot even on the same host and under the same temperature/humidity, it seems ‘risky” (?) to rely on N = 1 per combination. Is there a reason why a larger number of pairs wasn’t used? Perhaps provide explanation/justification.
201 – ‘determined the number of replicates used’ – somewhat vague. Can more specific language be used here?
213 – I am still not sure what sample sizes were used to produce Fig. 2 and Table 1 but believe this information needs to be presented.
235 – I feel a brief explanation of the data portrayal should be included in the Fig. 2 legend. Horizontal bars = means? Medians? Bars represent? Whiskers?
249-250 – same comments as for line 213 and line 235
262 – delete and before had
274-275 – according to Table 3, temperature had no significant effect on oviposition period duration, yet in text an inverse relation between temperature and oviposition duration is referred to. Am I missing something? Text seems to contradict Table 3.
370 – change to …larval durations and linked to…
412-424 – The test flies were provided unlimited food and water. Perhaps the authors should comment on the possibility that food (periods of starvation) and water (desiccation) stress might yield different results than those presented here for lifespan and fecundity. Regarding fecundity, for example, would food stress tend to have a greater inhibitory effect on females that were immatures under high temperatures or would food stress impact females equally from different immature thermal regimes?
Author Response
- Clarify issues of sample size (lines 186-196, line 201, line 213).
Response: Following the comment, we included a Supplementary Table S1, with number of individulas, and the number of replicates are given in 2.3 sections (lines 190-193)
- Line 37 – delete a
Response: Following the comment, we deleted “a” (line: 37)
- 38 – what are ecological patterns? Use more precise language.
Response: Following reviewers comment we explained what we mean (line 38).
- 40 – why both ecological and environmental effects? How can such effects be distinguished. Re-word.
Response: Thank you for the comment. We reworded in the revised text (line 40-41)
- 41 – delete the; change insects to insect
Response: Thank you for pointing out. In the revised text I have replaced insects with insect (line 42).
- 58 – insert comma after closing parenthesis; delete the before global
Response: Followinf the reviewers comment, we added a comma and deleted “the “in the revised text (line 58).
- 59 – change to Having originated in Africa, …
Response: Following reviewers comment we revised the text (line: 59)
- 70- replace at with to
Response: Thanks for pointing out. In the revised text we replaces at with to (line 71).
- 71 – delete comma after traits
Response: We deleted comma in the revised text (line 71)
- 72- change remain to remains [singular to agree with extent]
Response: We replaced remain with remains (line 72)
- 74 – insert comma after factors
Response: Thanks for pointing out, we added “,” after factors (line 74)
- 81 – delete (2024)?
Response: Thank you for your suggestion. However, we prefer to keep the year in the reference as it provides clarity regarding the specific publication date.
- 84 – delete comma [after …bitter oranges)]
Response: Following reviewers comment, we deletes the comma after the parenthesis (line 84).
- 86 – remain in orchards [not on]
Response: Thank you for pointing out. We replaced it, in the revised text (line 86)\
- 87 – delete plastic
Response: Following the comment, in the revised text we deleted plastic (line 87)
- 91 – insert comma after changes
Response: Following the comment, we added a “,” in the revised text (line 91).
- 92 – insert comma after stress
Response: Following the comment, we added a “,” in the revised text (line 91).
- 92 – delete as
Response: Following the comments we deleted “as” in the revised text (line 92)
- 103 – change to ..medfly’s life history characteristics, such as…
Response: Following the comment we replaced biological with life history (line 103)
- 120 – delete (1998)?
Response: Thank you for your suggestion. However, we prefer to keep the year in the reference as it provides clarity regarding the specific publication date.
- 122 – change to …regions, where host fruits are more likely to be available throughout the year…suggesting constancy in fruit availability is perhaps unwarranted.
Response: Thanks for suggestion, in the revised text we changed it as proposed (line 122).
- 125 – insert comma after traits
Response: Following reviewers comment a “,” was inserted (line 124)
- 133 – change effect to effects
Response: Thanks for pointing out. In the revised text we replaced effect with effects (line 133).
- 135 – what is a fecundity rate? Just use fecundity. Two hanging words = ‘plastic response’ – delete.
Response: Following reviewer comment we deleted rate. Although for comment on the plastic response, we added “as part of” accordingly to the comments of the 1st reviewers (line 135).
- 145 – insert and before Greece
Response: Thanks for pointing out, we added “and” as proposed (line 145)
- 146 – not sure what phrase ‘To avoid altitudinal clines’ means. Perhaps delete this.
Response: Altitudinal clines refer to variations in a biological trait or characteristic that occur along an altitude gradient, typically due to changes in environmental conditions with increasing or decreasing elevation. In our study, avoiding altitudinal clines ensures that observed variations in traits are more likely attributed to latitude (which correlates with climatic and environmental differences) rather than confounding effects of altitude. Thus, we prefer to keep this phrase in out text.
- 148 – change to The population from Croatia…
Response: Following the comment we revised the text accordingly (line 148)
- 150 – change to ...had been maintained…
Response: Following the comment we revised the text accordingly (line 150)
- 151-152 – Three of the capitatapopulations from Greece, those from Volos, Chios, and Chania, developed in oranges, while the population from Thessaloniki developed in apples.
Response: Following the comment we revised the text accordingly (line 151-153)
- 156-157 – change to …100 adults per cage (wooden, wire-mesh, 30x30x30 cm) at 25 + ….
Response: Following the comment we revised the text accordingly (line 158)
- 160 – delete mixture; so I understand – all ingredients were mixed together?
Response: Following the comment we revised the text accordingly. Yes all the ingredients were mixed together (line 161)
- 163 – change to trigger
Response: Thanks for pointing out. We revised it (line 1648)
- 165-168 – the generation numbers given refer to generations of rearing in lab prior to use in the experiments, correct? Perhaps state that explicitly.
Response: Yes the generations timed refer to generations of rearing in the lab before using them for an experiment. For clarity we added a sentence at the end of the paragraph (lines 166).
- 2 – reverse order- use Host Fruits as section heading
Response: Following reviewers comment, we reversed the order (line 171)
- 178 – delete (2024)?
Response: Thank you for your suggestion. However, we prefer to keep the year in the reference as it provides clarity regarding the specific publication date.
- 180 – delete comma after closing parenthesis
Response: Following the comment we deleted the “,” in the revised text (line 182)
- 184 – how many fruit of each type were placed in each of the 3 temperature regimes?
Response: To clarify this we added a sentence with the exact replicates (line 190-193)
- 184 – change to artificially
Response: Thanks for clarifying. We revised it (line 186)
- 186-189 – put sample size information above; delete sentence starting This experimental setup…
Response: To clarify this we added a sentence with the exact replicates (line 190-193)
- 193-196 – Do I understand correctly? Only 1 pair of flies was used per origin/host/temperature? As female fecundity can vary a lot even on the same host and under the same temperature/humidity, it seems ‘risky” (?) to rely on N = 1 per combination. Is there a reason why a larger number of pairs wasn’t used? Perhaps provide explanation/justification.
Response: Thank you for your observation. To clarify, more than one pair of flies was used for each combination of origin, host, and temperature. All adults that emerged from each treatment were paired (one male and one female per pair), ensuring sufficient replication across the treatments to account for variability in female fecundity and other traits. We revised the text to make this point clearer (lines: 197-199)
- 201 – ‘determined the number of replicates used’ – somewhat vague. Can more specific language be used here?
Response: Thank you for your question. The number of pairs used in each treatment was determined by the number of adults that emerged, which varied across treatments due to natural differences in emergence rates. While this resulted in unequal sample sizes, we accounted for this variability by using statistical methods appropriate for unbalanced designs. Additionally, we ensured that all available replicates were included to maximize the robustness of our analyses. To clarify this we revised the text (lines 205-207)
- 213 – I am still not sure what sample sizes were used to produce Fig. 2 and Table 1 but believe this information needs to be presented.
Response: We added Supplementary Table S1.
- 235 – I feel a brief explanation of the data portrayal should be included in the Fig. 2 legend. Horizontal bars = means? Medians? Bars represent? Whiskers?
Response: Thank you for your comment. The boxplots in Figure 2 include the median, the 1st quartile (25th percentile), and the 3rd quartile (75th percentile). The whiskers represent the lowest and highest values within the interval defined by ±1.5 times the interquartile range (IQR) from the 1st and 3rd quartiles. Values outside this range are considered outliers and are depicted as individual points. We added in the legend (line 243-245)
- 249-250 – same comments as for line 213 and line 235
Response: We revised the text accordingly to the comments.
- 262 – delete and before had
Response: Thanks for pointing out this. We have deleted (line 273)
- 274-275 – according to Table 3, temperature had no significant effect on oviposition period duration, yet in text an inverse relation between temperature and oviposition duration is referred to. Am I missing something? Text seems to contradict Table 3.
Response: Thank you for pointing that out. The text refers to the interaction between population and treatment, which had a significant effect on oviposition period duration, rather than a direct relationship between temperature and oviposition. While temperature alone did not show a significant effect, the interaction between population and treatment, including temperature, did influence oviposition period duration. We revised the text to be more specific (line 421-422).
- 370 – change to …larval durations and linked to…
Response: Thank you for your suggestion. In response to another reviewer’s request to avoid repetition of results, we decided to remove this phrase in the revised text. We felt that the information was already conveyed earlier in the manuscript, and its removal helped streamline the presentation (line 406-407)
- 412-424 – The test flies were provided unlimited food and water. Perhaps the authors should comment on the possibility that food (periods of starvation) and water (desiccation) stress might yield different results than those presented here for lifespan and fecundity. Regarding fecundity, for example, would food stress tend to have a greater inhibitory effect on females that were immatures under high temperatures or would food stress impact females equally from different immature thermal regimes?
Response: Thank you for your thoughtful comment. We revised the text accordingly (line 428-432).